# Analysis of Head Micromovements and Body Posture for Vigilance Decrement Assessment

Dario Rossi [1], Pietro Aricò [2,3], Gianluca Di Flumeri [1,2], Vincenzo Ronca [2,3], Andrea Giorgi [4], Alessia Vozzi [4], Rossella Capotorto [4], Bianca M. S. Inguscio [1,2], Giulia Cartocci [1,2], Fabio Babiloni [2,5,6,*] and Gianluca Borghini [1,2]

[1] Department of Molecular Medicine, Sapienza University of Rome, Viale Regina Elena, 291, 00161 Rome, Italy; dario.rossi@uniroma1.it (D.R.); gianluca.diflumeri@uniroma1.it (G.D.F.); biancams.inguscio@uniroma1.it (B.M.S.I.); giulia.cartocci@uniroma1.it (G.C.); gianluca.borghini@uniroma1.it (G.B.)

[2] BrainSigns srl, Via Tirso, 14, 00198 Rome, Italy; pietro.arico@uniroma1.it (P.A.); vincenzo.ronca@uniroma1.it (V.R.)

[3] Department of Computer, Control, and Management Engineering "Antonio Ruberti", Sapienza University of Rome, Piazzale Aldo Moro 5, 00185 Rome, Italy

[4] Department of Anatomical, Histological, Forensic & Orthopedic Sciences, Sapienza University of Rome, Piazzale Aldo Moro 5, 00185 Rome, Italy; andrea.giorgi@uniroma1.it (A.G.); alessia.vozzi@uniroma1.it (A.V.); rossella.capotorto@uniroma1.it (R.C.)

[5] Department of Physiology and Pharmacology "Vittorio Erspamer", Sapienza University of Rome, Piazzale Aldo Moro 5, 00185 Rome, Italy

[6] Department of Computer Science, Hangzhou Dianzi University, Hangzhou 310018, China

[*] Correspondence: fabio.babiloni@uniroma1.it; Tel.: +39-3287697914

**Abstract:** Vigilance refers to the capability of humans to respond accordingly to relevant and unpredictable tasks and surrounding environment changes over prolonged periods of time. Identifying vigilance decrements can, therefore, have huge and vital impacts on several operational environments in which a simple slip of mind or a deficit in attention can bear life-threatening and disastrous consequences. Several methodologies have been proposed to assess and characterize vigilance, and the results have indicated that the sole measure of performance and self-reports are not enough to obtain reliable and real-time vigilance measure. Nowadays, monitoring head and body movements to obtain information about performance in daily activities, health conditions, and mental states has become very simple and cheap due to the miniaturization of inertial measurement units and their widespread integration into common electronic devices (e.g., smart glasses, smartwatches). The present study aimed to understand the relationship between head micromovements and body posture changes to vigilance decrease while performing the psychomotor vigilance task. The results highlighted that head micromovements can be employed to track vigilance decrement during prolonged periods of time and discriminate between conditions of high or low vigilance.

**Keywords:** vigilance; inertial measurement units; psychomotor vigilance task; head micromovements; body posture

## 1. Introduction

A vast corpus of studies has highlighted that cognitive processing (e.g., visuospatial ability, memory, attention and executive functions) appears to be influenced by the contribution of the vestibular system (for a review see [1]). Typically, the role of this system is to maintain gaze stability and body position and stabilize head movements, but attention-demanding tasks have consistently shown a decrease in performance (e.g., response latency, accuracy) when the vestibular system is challenged [2–4]. In particular, the results showed that body posture did not worsen when cognitive tasks were added, indicating that the brain prioritizes balance and posture stability. This indicates that cognitive tasks are not

simply reflexive but compete with attention for cognitive resources [1]. This evidence suggests that vigilance and the vestibular response (in term of head movements and body posture) could require processing functions by similar cognitive networks.

The definition of vigilance in the scientific literature is ambiguous. In fact, it varies according to the field that is being studied. For example, in psychology and cognitive neuroscience, vigilance is described as the ability of the observer to sustain attention over a prolonged period of time under monotonous stimulus [5]. The vigilance level is a concept to which clinical neurophysiologists usually refer when describing the activity of the corticothalamic networks implied in the sleep–wake dimension [6]. Vigilance has also been defined through the degradation of performance results over time while involved in cognitively simple tasks [7], indicating the involvement of time as a factor to define and/or measure it.

Common definitions of vigilance usually contain terms closely related to arousal, alertness, and sustained attention. Arousal refers to a nonspecific activation of the cerebral cortex and the neurobiological mechanism behind vigilance itself, with low levels related to sleep and high levels related to the vigilant state [8]. In a hyper-aroused condition, it is also associated with models describing insomnia [9]. Alertness has been described as the state of maintained high sensitivity to incoming stimuli [10] or the quantitative measure of the state of the mind to being sensitive to internal or external stimuli [8]. Attention is one of the basic human cognitive abilities, allowing for the discrimination of relevant parts of information and the ability to discard the others. Attention is related to a focused activation of cerebral cortex that enhances information processing [11]. By extension, the concept of sustained attention is the ability to maintain a mental state of focused attention and alertness over time [12], a definition that is very close to that of vigilance, which is typically used as a synonym [13].

Most vigilance definitions refer to the capability of humans to respond accordingly to relevant and unpredictable changes over prolonged periods of time while dealing with tasks. We have, therefore, considered this definition for the proposed study [14,15].

Vigilance can also be influenced by cognitive processing, motivation, and stress [16]. An important factor to motivation is the dopamine system related to the reward [17], indicating that performance and vigilance can decrease in non-stimulating environments. Stress also has impacts on vigilance and relative associated performance [18], and several neurophysiological markers associated to stress (e.g., EEG, skin temperature, electrodermal activity, heart rate variability, blood pressure and breathing) revealed that stress and cognitive functions have a U-shaped curve relationship. This suggests that opportune levels of stress can, in fact, improve vigilance-related performance [16].

Although, over the last 50 years, automation technology has profoundly changed human–machine interactions (HMI), high levels of automation can have negative consequences due to, for example, excessive trust in autonomous systems' abilities [19] or the loss of situational awareness [20], which causes the well-known out-of-the-loop phenomenon (OOTL) [21,22]. This leads to a decrease in operator vigilance and contributes to the failure to detect and understand the problem and make the right decision. A conspicuous number of accidents caused by vigilance decrement, in particular in the aviation field [23,24] or during vehicle driving [25,26], has been widely recognized. For example, a recent study by Greenlee et al. [27] highlighted the importance of monitoring vigilance in drivers engaged with automated vehicles. The results showed that the drivers experienced a decrease in sensitivity to hazards and an increase in false alarms in the automated control condition in respect to the manual control condition. Because the presence on the streets of automated or partially automated vehicles for consumers' use is projected to increase, the importance of identifying and tracking states of low vigilance appears to be crucial. Thus, the capability of identifying vigilance degradations can have many benefits in all contexts in which a simple slip of mind or a deficit in attention can bear life-threatening and disastrous consequences.

Several methodologies and markers have been proposed to characterize and assess vigilance changes. The psychomotor vigilance task (PVT) [28] is a reliable and widely used

method to monitor users' vigilance over a prolonged period of time (at least 10 min). PVT measures vigilance degradation by recording reaction times to visual or auditory stimuli that occur at random intervals (typically ranging from 1 to 10 s). In the PVT, cognitive impairments due to vigilance degradation is reflected not only by the identification and response to the target stimuli (i.e., reaction time) but also by the number of missed targets and the false response in the case that no stimulus is presented. Moreover, subjective measures, such as the visual analogue scale (VAS) [29], have also been used to rate perceived vigilance level. Significant differences in vigilance ratings were observed between participants, and the results were not consistent for different tasks [30,31]. These results indicate that the sole measure of performance and self-reported questionnaires are not enough to have generalizable results on vigilance [32]. Most importantly, questionnaires cannot be acquired during the execution of a task, with the drawback of compromising the accuracy and reliability of the measure they intend to evaluate [33,34]. However, performance data, although available during the execution of the task, are strongly related to the task at hand, so it is difficult to compare results obtained from different settings [35].

Data collected using inertial measurement units (IMUs), such as head micromovements and body posture changes, and neurophysiological measures, such as electroencephalography signal (EEG), can allow us to overcome the drawbacks of subjective measures by objectively assessing the user's cognitive states in real time during the execution of a task. The application of EEG and its reliability has been already well explored by the authors in a variety of laboratory and realistic settings by evaluating human–robot interaction in training assessments, driving, and air traffic control [36–40]. Although commercial and cheap EEG devices are available on the market, their correct usage requires specialized personnel to be able to check the correct position of the sensors (electrodes located exactly over the corresponding brain area) and quality of the EEG signals to achieve the results [41,42]. Therefore, the possibility of monitoring vigilance degradations through the analysis of head and body movements could be a valid alternative, especially due to the simple and cheap integration of IMUs (accelerometers, gyroscopes) into electronic devices such as smartwatches, virtual reality, and biosignal recording systems (e.g., EEG). This aspect also allows for tracking movements in environments in which global positioning system (GPS) tracking is not possible or difficult [43,44]. IMUs have been extensively used in clinical applications to monitor patients' rehabilitation both in conditions in which robotic exoskeletons are involved and during free movement rehabilitation [45], stroke rehabilitation [46], and posture evaluation and rehabilitation progression in children with cerebral palsy [47,48]. Also, in operational environments, the recognition of human activity (HA) by the means of wearable sensors has gained high importance to safely assess the position in time and space of the operators and improve their performance, especially where human–robot interaction is involved [49]. For example, Ramirez et al. [50] used inertial sensors to spot the visual focus of attention of a driver, while Lee et al. [51] embedded an inertial sensor in a custom-made glove to assess driver stress based on driving behavior.

Previous studies have highlighted how it is possible to identify different human activities [52], discriminate stress conditions [51] or variations in vigilance and drowsiness [53] based on drivers' steering behavior, or identify different positions of the head with the possibility of linking them to different attentional states [54] with the use of inertial sensors. However, to the best of our knowledge, the micromovements of the head and variations in body posture with a decrease in vigilance have not yet been addressed. The present study, therefore, aims to understand the relationship between the micromovements of the head and changes in body posture with vigilance decrease by analyzing data from inertial sensors. In particular, given the potential of IMU devices, the present study aims to develop and validate a vigilance index based on the user's head micromovements and body posture. In fact, data collected through IMUs are easily available and do not require professional personnel for setting up the sensors on the user's body, as the neurophysiological measures do.

## 2. Materials and Methods

### 2.1. Sample Population

Thirteen healthy participants (27 ± 3 years old, 7 males and 6 females) were enrolled on a voluntary basis in this study. The selection of the participants has been performed accurately to ensure the same mental and physical status (homogeneity of the experimental sample). They have been asked about past neurological and physical disorders and instructed to maintain a specific kind of lifestyle. For example, they have been asked to avoid alcohol, heavy meals, and caffeine right before the experiments (homogeneity of the "internal conditions" of the subjects during the experiments). The lab environment has been kept under control (lights intensity, room temperature, seat position) across the different days of the experiments (homogeneity of the "external conditions" during the experiment). Written informed consent was obtained from each participant after the explanation of this study and before the start of the data acquisition. The experiment was conducted following the principles outlined in the Declaration of Helsinki of 1975, as revised in 2008. It received a favorable opinion from the Ethical Committee of Fondazione Santa Lucia (Prot. CE/PROG.604 dated 5 April 2017). Moreover, the participants were informed on how to complete the tasks proposed later during the experimentation, and all of them took part in a practice session before starting with the experiment to avoid compromised results due to learning and familiarization effects. Then, a resting phase was considered before the start of the actual experiment session. Due to missing data, one participant has been excluded from the analysis. Thus, the final sample population is composed of 12 participants completely balanced between males and females.

### 2.2. Psychomotor Vigilance Task (PVT)

All the participants performed two separated PVT with a conjunction visual search task in between (not considered in this study, as it is specific for selective attention functions). After each task, a resting period was considered according to the participants' disposition to avoid causing visual strain that could confound and impair the correct evaluation of vigilance decrease. Therefore, the entire protocol consisted of three phases, with a total duration of about 45 min and each phase lasting a maximum of 15 min. Moreover, screen distance, luminance, and contrast were adjusted according to participant's demands [55,56]. The PVT, which is a specific test to induce vigilance degradation, in this study consisted of 10 min of continuous stimuli presentation on a monitor with random *interstimulus intervals* (ISI) ranging from 1 to 10 s. The duration of the PVT was set based on the results obtained from Loh et al. [57], in which a significant vigilance degradation was observed after 10 min. The participants had to press the space bar on a keyboard in front of them as fast as possible in response to the stimuli presentation (a red circle in the center of the screen) after the appearance of a fixation cross (Figure 1). Thus, the PVT was composed of multiple subsequent repetitions of trials, which included (1) ISI; (2) a fixation cross; and (3) target stimulus. During the entire protocol, the participants were seated on a comfortable chair in front of a computer screen. Moreover, high-resolution electroencephalography (HR-EEG) signal was acquired using a 61-channel system (see Sebastiani et al. [58] for more details on data acquisition and analysis), and micro and macro movements were recorded using 2 IMU devices composed of a 3-axis accelerometer and a 3-axis gyroscope placed on the chest and forehead of the participants. Moreover, the participants' reaction time (RT) in response to the correct target stimuli was collected to measure participants' performance. The RT was obtained by timing the time between the target onset (red dot) and the participant hit on the keyboard. Participants' EEG signals, IMU data and on-screen stimuli were synchronized for the entire duration of the protocol. Before the beginning of the experiment, one minute of resting state with open eyes in front of the blank monitor (OA) was recorded to obtain head micromovements and body posture measures from a *movement-free* baseline condition.

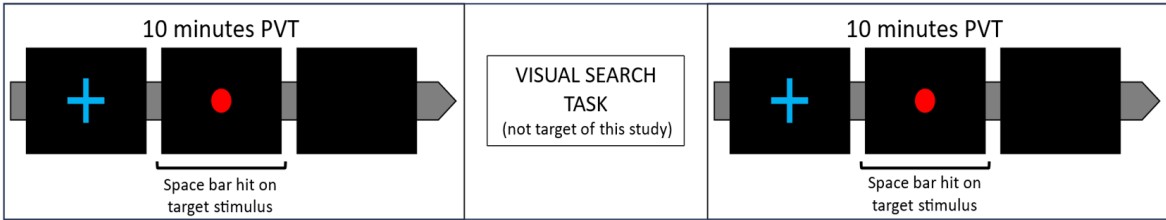

**Figure 1.** Graphical representation of stimuli presented to participants and the flow of the protocol proposed, consisting of two separate psychomotor vigilance tasks (PVT) and a visual search task in between. Resting periods have been considered at the end of each task to avoid visual strain. The PVT required the participants to press the space bar as fast as possible after the onset of a red dot which appears on the screen after a blue fixation cross.

### 2.3. Acceleration Data Recording and Processing

Head and chest acceleration were recorded by using two Shimmer GSR3+ systems (Shimmer Sensing, Dublin, Ireland) with integrated inertial sensing via accelerometer and gyroscope at a sampling rate of 100 Hz. Before the beginning of each experimental session, the accelerometer and gyroscope of the IMU devices were calibrated on a flat and stable surface according to manufacturer guidelines to obtain their relative correct offsets, sensitivity, and alignment matrices. Once the calibration was completed, we positioned one device on the forehead and one on the chest of each participant. Then, from each device, the linear acceleration was obtained through data fusion of the acceleration and angular velocity with the use of the Madgwick filter [59] implemented in the *imufusion* Python package. Finally, we used the linear acceleration from the three axes to calculate the modulus of the acceleration of the head and the chest. To describe the data distribution related to possible micromovements associated to vigilance, for each minute of the PVT, the median value and the median absolute deviation (MAD) of the modulus of the accelerations were estimated as a measure of the intensity and variability of participants movements, respectively. In other words, the median value of the acceleration describes the magnitude (i.e., how much) over time of the participants' head micromovements and body postures changes, while the MAD describes the variation over time in the magnitude exhibited by the participants' head micromovements and body posture variations due to vigilance variations. Additionally, the use of statistical parameters like the median and MAD allowed us to obtain distributions that were not heavily influenced by possible outliers due to sporadic large movement of the head and the body. In particular, values derived from the sensor on the head were used as estimations of head micromovements because they are related to attentional states [54], while the sensors on the chest were used as estimations of body posture changes (for a review see [60]). Acceleration median and MAD values were then normalized with respect to the corresponding values obtained during the movement-free condition (open eyed phase) by using median and median absolute deviation (MMAD) normalization due to its robustness to outliers [61].

### 2.4. Behavioral Data: Reaction Time

During the entire PVT, participants' reaction times (RT) in response to each trial were recorded. RTs were defined as the time elapsed from the onset of target stimulus to the spacebar press. For each minute of the PVT, only the RTs related to correct answers and trials within the corresponding minute were averaged.

### 2.5. Statistical Analyses

Participants' RTs, acceleration median, and MAD of head micromovements and body posture of both PVTs were averaged. Then, trends over time of these parameters were analyzed using Page's trend test [62] to assess the significance of the trends over time to understand if there was a decrease or increase in those parameters from the beginning to the end of PVT.

The Wilcoxon signed rank test was used [63] to confirm any vigilance decrement between the first and last minute of the PVT of the parameters with statistically significant trends. Statistical analyses were performed using Python *scipy* [64] and *pingouin* [65] packages.

### 3. Results

*3.1. High- and Low-Vigilance Conditions*

First, we performed the Page's trend test to verify a possible decline of vigilance over time in terms of behavioral data (RT). The results confirmed a statistically significant increase in RT from the first to the last minute of the PVT (L = 4059, $p < 0.001$). In other words, this result indicates that the participants experienced a vigilance decrement while performing the PVT (Figure 2A). Moreover, we wanted to find out whether such a vigilance decrement between the beginning and end of the PVT was significant. In this regard, the Wilcoxon signed rank test reported a statistically significant increase (W = 63, $p = 0.03$) of the participants' RT between the first and last minute of the PVT (Figure 2B). Vigilance decreases over time (Page's trend test results: L = 4271, $p < 0.01$) and, between the first and last minute of the task, was also confirmed using the EEG–based vigilance index (Wilcoxon signed rank test results: W = 77, $p < 0.01$), which was calculated as proposed by the authors in a previous study [58]. The results obtained from behavioral data and the vigilance index confirmed the presence of vigilance degradation between the first (high-vigilance condition) and the last minute of PVT (low-vigilance condition).

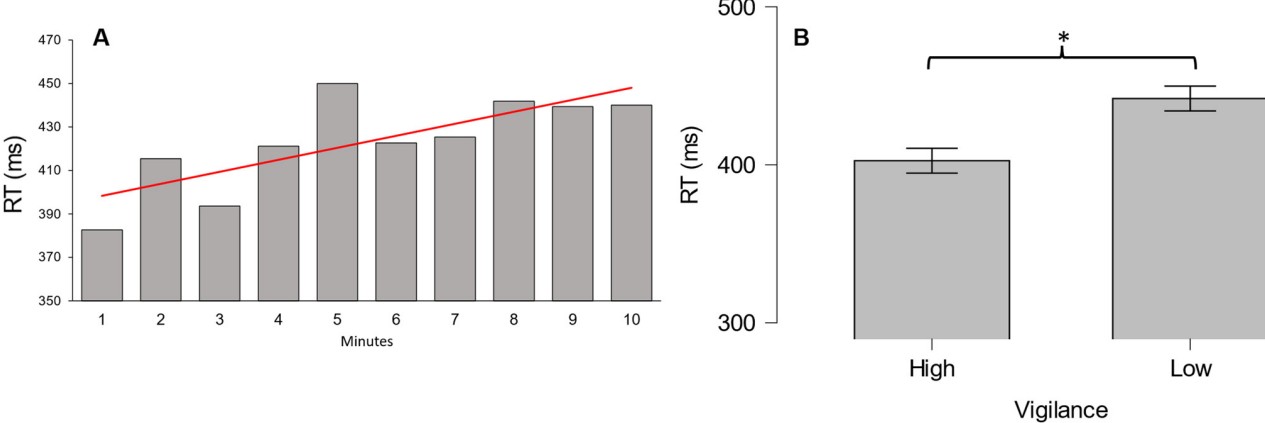

**Figure 2.** (**A**) Vigilance decrement over time during PVT indicated by RT increase; (**B**) RT increase in low-vigilance condition. * denotes $p < 0.05$. Red line indicates significant data trend.

*3.2. Acceleration Results: Head Micromovements*

The results for the head micromovements showed a statistically significant positive trend over time both for the median (intensity) and the MAD (variability) of the acceleration (L = 4041, $p < 0.01$ and L = 3866, $p = 0.01$, respectively) during the execution of the PVT (Figures 3A and 4A). Figures 3B and 4B show that there was a significant increase in median and MAD acceleration of head micromovements (W = 6, $p < 0.01$ and W = 13, $p = 0.02$, respectively) when the vigilance decreased significantly.

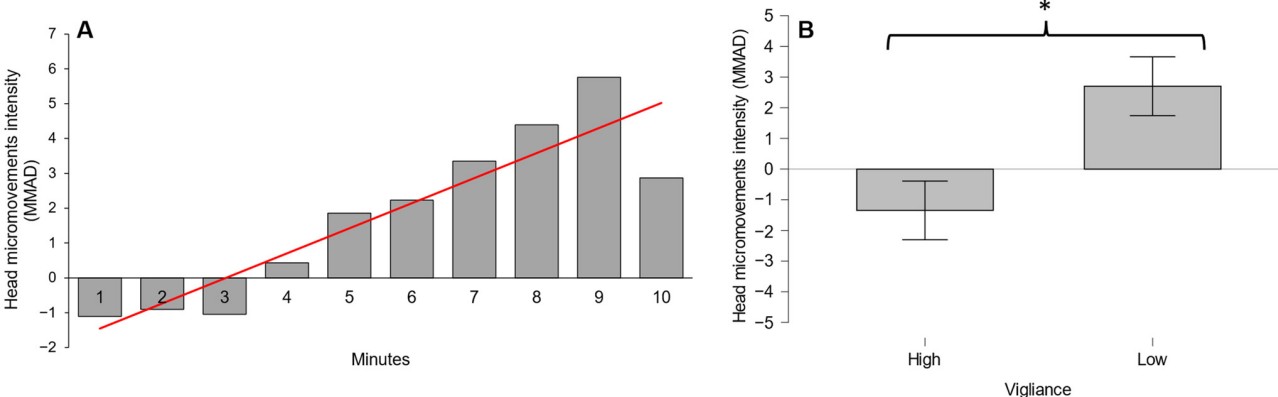

**Figure 3.** (**A**) Head micromovement intensity increase over time during PVT; (**B**) Head micromovement intensity increase in low-vigilance condition. * denotes $p < 0.05$. Red lines indicate significant data trends.

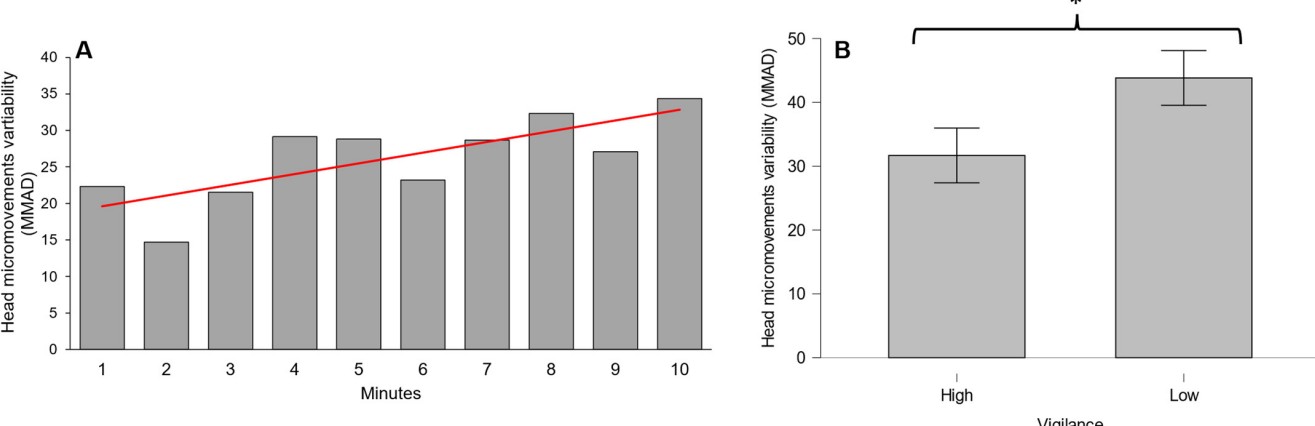

**Figure 4.** (**A**) Head micromovement variability increase over time during PVT; (**B**) Head micromovement variability increase in low-vigilance condition. * denotes $p < 0.05$. Red lines indicate significant data trends.

### 3.3. Acceleration Results: Body Posture

The results for the body posture showed a statistically significant positive trend over time ($L = 3881$, $p < 0.01$) for the median (intensity), but no significant trend ($p = 0.32$) was found for the MAD (variability). In addition, the statistical analysis between the beginning and the end of the PVT did not return any significant changes for the median or MAD acceleration of the body posture ($p = 0.08$ and $p = 0.28$, respectively). For this reason, body posture was not considered as a possible marker of vigilance decrease in the subsequent analysis.

### 3.4. Repeated Measures Correlations

Based on the results derived by head micromovements data, we performed a repeated measures correlation analysis to better understand the relationships between RT and head micromovements. We calculated a moving root mean square (rms) over time for each of the two variables analyzed. The results indicated a weak positive statistically significant correlation between RT and acceleration measures with an $r = 0.27$ ($p < 0.01$) for head micromovements intensity (Figure 5A), while showed a strong positive statistically significant correlation with an $r = 0.72$ ($p < 0.01$) for variability (Figure 5B).

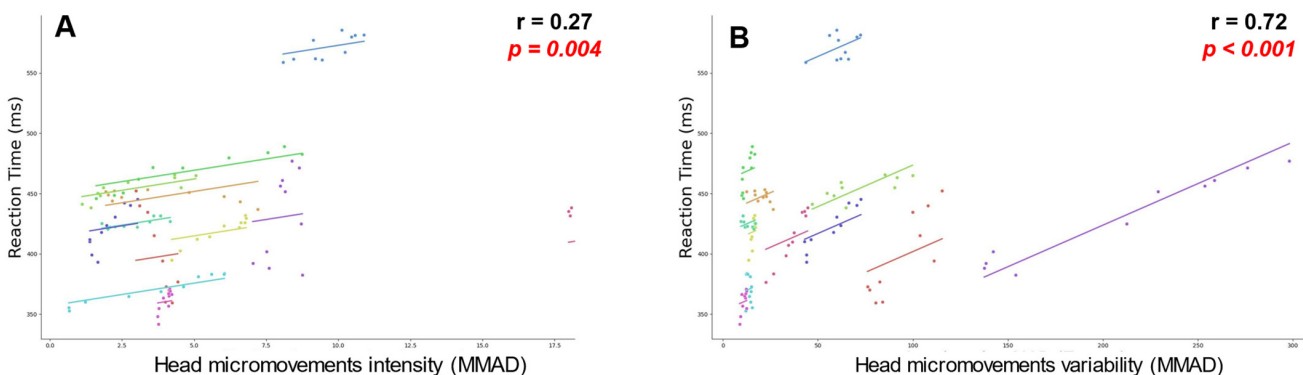

**Figure 5.** Repeated measures scatterplots (subject data in different colors). (**A**) Correlation between Reaction Times and head micromovements intensity; (**B**) Correlation between Reaction Times and head micromovements variability. All correlations are significant with $p < 0.05$, highlighted in red in the figure.

## 4. Discussion

The results obtained in this study confirmed the possibility of assessing vigilance degradations through the analysis of participants' head micromovements. A significant vigilance reduction at the end of the PVTs [57] was highlighted by participants' reaction times (RTs) and also confirmed by a previous study [58] (Figure 2). In this regard, participants' head micromovements were able to discriminate low- and high-vigilance conditions. In fact, when the vigilance decreased significantly, we found increased intensity and variability of micromovement acceleration over time (Figures 3 and 4). Meanwhile, the results derived from body postures showed that, although there was a positive trend for acceleration intensity, we did not find any significant differences between the high- and low-vigilance conditions, both for acceleration intensity and variability.

Moreover, the analysis of head micromovement variability showed a positive and significant correlation with participants' RT (Figure 5), indicating that this parameter could be used to assess the current state of operators' vigilance while dealing with tasks.

Taken together, the results hint at the possible implication of the vestibular system with the vigilance assessment. In particular, it seems that in a state of high vigilance, the cognitive resources are sufficiently balanced between PVT execution and the request of the head and the body to maintain their position, which is most likely in an apparent nonconscious way. Meanwhile, the low-vigilance condition leads to higher cognitive demand to hold the head steady, leading to an increase in conscious control of head micromovements, while body position is not affected. Lower resources are instead dedicated to PVT execution, leading to a decrease in performance, as highlighted by the increase in RT. As a matter of fact, previous studies have pointed out that, when the vestibular system is challenged, maintaining the same balance and posture starts to be cognitively demanding [1,2,66]. However, on the other hand, the incorporation of vestibular stimulation, like swinging on a swing during leisure time, as suggested in [67], over time could instead improve cognitive function. Although plausible, the link between vigilance and the vestibular system is far from being resolved by the present study. More focused studies should address the actual causal relationship between the two.

The results of the present study also demonstrate that IMUs endow a cheap, direct, and reliable measure of participants' vigilance levels. Due to the integration of IMUs with neurophysiological devices, these kinds of data could also be combined for assessing vigilance changes in terms of participants' behavior and cognitive response. This integration could help to identify and, if necessary, alert operators of conditions of vigilance deterioration. There are plenty of operation environments in which it could be possible to integrate inertial measurements on already existing equipment, such as in aviation on traffic controller operator or pilots' headphones or in environments in which security helmets are used, for example, on construction sites for heavy machinery operators or

quality control. In this way, no additional device that could modify or impede a normal day-to-day work routine would be required, and the ergonomics of the existing ones would not be compromised, allowing researchers to overcome a possible barrier to the adoption of such devices. The outcome can be used to create a closed loop between the user and the machine and make them continuously interact to mitigate OOTL phenomena and improve both users' performance and task safety.

Moreover, the measure that we proposed here in this study does not require extensive computational time ($14.45 \pm 1.79$ ms is the average time for the elaboration of 60 s of data with an Apple M1 CPU), allowing for online vigilance monitoring. This means that, with proper calibration on a movement-free phase, the index could be also implemented as a direct measure that can be used to adapt user interfaces when vigilance is lowered below a predefined threshold [68].

Despite the innovative and interesting results, some limitations must be discussed. First, the sample size consisted of 12 participants within a narrow age range, and those factors could be a limit to a broader generalization of the findings reported in this study. Therefore, in future studies, we will enlarge the population and include different age ranges to provide more reliable evidence and substantiate the findings reported in this study. Secondly, we want to estimate the time resolution by which the IMUs can assess vigilance decrement. In this regard, we need a task that is able to provide performance data with high temporal resolution (e.g., every second) so that we will be able to identify vigilance degradation with a resolution of seconds and perform correlations between participants' performance and other kinds of data (i.e., IMUs). Finally, future studies on head micromovements related to vigilance decrease will have to take into account the role of motivation and stress and their effect on the possible degradation of performance and how they affect vigilance [17,18].

## 5. Conclusions

Given the miniaturization of inertial sensors, the use of IMUs can be easily implemented in all operating environments in which is crucial to adequately evaluate vigilance degradation and prevent accidents related to it [24,69], such as in aviation [36], while driving a vehicle [70], or in a surgery room [71]. The capability of using head micromovements, as described in this study, paves the way to extending this strategy to track other kinds of mental states (e.g., mental workload, stress) by exploiting technological progress and the integration of IMUs in commercial and personal devices.

**Author Contributions:** Conceptualization, D.R. and G.B.; Data curation, D.R., B.M.S.I. and G.B.; Formal analysis, D.R., P.A., G.D.F., B.M.S.I. and G.B.; Funding acquisition, F.B. and G.B.; Investigation, D.R. and G.B.; Methodology, D.R., P.A., G.D.F., V.R., A.G., A.V., R.C., G.C. and G.B.; Project administration, F.B. and G.B.; Resources, D.R. and G.B.; Supervision, F.B. and G.B.; Visualization, D.R., B.M.S.I., G.C. and G.B.; Writing—original draft, D.R.; Writing—review and editing, D.R., P.A., G.D.F., V.R., A.G., A.V., R.C., B.M.S.I., G.C. and G.B. All authors have read and agreed to the published version of the manuscript.

**Funding:** This work was financed by the European Commission by Horizon 2020 projects Sesar-01-2015 Automation in ATM, "STRESS" (GA No. 699381).

**Institutional Review Board Statement:** The experiments were conducted following the principles outlined in the Declaration of Helsinki of 1975, as revised in 2013. The experiments have been approved by the Ethical Committee of Fondazione Santa Lucia (Prot. CE/PROG.604 dated 5 April 2017).

**Informed Consent Statement:** Informed consent was obtained from all subjects involved in this study.

**Data Availability Statement:** The data presented in this study are available on request from the corresponding author. The data are not publicly available due to privacy reasons.

**Acknowledgments:** This work was co-funded by the individual grants "CHALLENGES: CompreHensive frAmework to aLLEge and analyze surGEons' Stress" (Bando Ateneo Medio 2021), "BRAINORCHESTRA: Multimodal teamwork assessment through hyperscanning technique" (Bando Ateneo Medio 2022) provided by Sapienza University of Rome to Gianluca Borghini, "AI-DRIVE: AI-based multimodal evaluation of car drivers' performance for onboard assistive systems" (Avvio alla ricerca 2021), provided by Sapienza University of Rome to Gianluca Di Flumeri, "The Smelling Brain: discovering the unconscious effect of the odours in industrial contexts" (Avvio alla ricerca 2022), provided by Sapienza University of Rome to Alessia Vozzi, "REMES–Remote tool for emotional states evaluation" provided to Vincenzo Ronca, and "HF AUX-Aviation: Advanced tool for Human Factors evaluation for the AUXiliary systems assessment in Aviation" provided by Sapienza University of Rome to Vincenzo Ronca.

**Conflicts of Interest:** All authors declare that the research was conducted in the absence of any commercial or financial relationships that could be construed as a potential conflict of interest.

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
