# Peer review of "Analysis of Head Micromovements and Body Posture for Vigilance Decrement Assessment"

_applsci, doi:10.3390/app14051810_

Round 1
Reviewer 1 Report
Comments and Suggestions for Authors
The authors explored the possibilities of analysing head micromovements and body posture for vigilance decrement assessment.
I have a few queries below related to the study:
1. Since the experiment lasted for 45 min, were there any notable issues related to visual fatigue? This can reduce the level of alertness.
2. For the reaction time, was the identification based on event-related potentials? P300?
3. Were the IMU data streams synchronised with the EEG data?
It will be helpful if a graphical representation of the experiment is included.
Reviewer 2 Report
Comments and Suggestions for Authors
This article titled "Analysis of Head Micromovements and Body Posture for Vigilance Decrement Assessment" presents an innovative study that investigates the correlation between head micromovements, body posture changes, and vigilance levels during the execution of the Psychomotor Vigilance Task (PVT). The authors aimed to develop and validate a Vigilance Index based on inertial measurement units (IMUs) data to offer a non-invasive, real-time, and objective measure of vigilance, particularly useful in operational environments where vigilance decrement could have critical consequences. Here are some of my comments for further enhancing the quality of the article:
1. The study's sample size is relatively small (12 participants), which might limit the generalizability of the findings. Increasing the sample size and including participants with diverse backgrounds could enhance the study's robustness and applicability.
2. The methodology must ensure rigorous calibration of IMUs to avoid data inaccuracies that could influence the Vigilance Index's reliability. Details on calibration procedures and error handling would strengthen this aspect.
3. The methodology involves selecting specific features from the IMU data to predict vigilance levels. A more detailed justification for the chosen features, including statistical tests or machine learning models used for feature selection, would enhance the methodology's transparency.
4. Implementing the Vigilance Index in operational settings requires real-time data processing capabilities. The article should discuss the computational requirements and the feasibility of real-time implementation in detail.
5. The methodology section is well-detailed, especially regarding the use of IMUs and statistical analyses. However, it could benefit from a more thorough explanation of the selection criteria for participants and the rationale behind the specific IMU placement locations.
6. The study intersects several disciplines, including cognitive neuroscience, psychology, and human-computer interaction. Expanding on the interdisciplinary implications of the findings and how they can be integrated into existing theories and practices across these fields would be beneficial.
7. The use of IMUs is innovative, yet the article could further elaborate on the practical aspects of implementing such technology in real-world settings. Discussing the feasibility, cost-effectiveness, and user acceptance could enhance the article's relevance to practitioners and researchers alike.
Round 2
Reviewer 2 Report
Comments and Suggestions for Authors
The authors have significantly addressed all the concerns. I recommend accepting the article in the current form.